# Household food insecurity and its association with self-reported male perpetration of intimate partner violence: a survey of two districts in central and western Uganda

George Awungafac ,[1] Stephen Mugamba,[2] Fred Nalugoda,[2] Carl Fredrik Sjöland ,[1] Godfrey Kigozi,[2] Susanne Rautiainen,[1] Robert Bulamba Malyabe,[2] Leo Ziegel,[1] Gertrude Nakigozi,[2] Grace Kigozi Nalwoga,[2] Emmanuel Kyasanku,[2] James Nkale,[2] Stephen Watya,[2] Anna Mia Ekström,[1] Anna Kågesten[1]

► Prepublication history and additional materials for this paper are available online. To view these files, please visit the journal online (http://dx.doi. org/10.1136/bmjopen-2020- 045427).

[1]Department of Global Public Health, Karolinska Institute, Stockholm, Sweden
[2]Africa Medical and Behavioural Science Organization, Kampala, Uganda

**Correspondence to**
Dr George Awungafac; awungafacg@yahoo.com

## ABSTRACT

**Objectives** This study aimed to determine the lifetime prevalence of male-perpetrated intimate partner violence (IPV), and to assess the association with food insecurity, sociodemographic factors and health risk behaviours in Uganda in the year preceding COVID-19-associated lockdowns.

**Design** Population-based, cross-sectional household survey.

**Setting** Urban, semiurban and rural communities of the Wakiso and Hoima districts in Uganda.

**Participants** A total of N=2014 males aged 13–80 years participated in the survey. The current study included males who reported having ever been in a sexual union and responded to the IPV questions (N=1314).

**Measures** Data were collected face-to-face from May 2018 to July 2019 using an interviewer-mediated questionnaire. Lifetime IPV perpetration was measured as 'no physical and/or sexual IPV', 'physical' versus 'sexual violence only', and 'physical and sexual violence'. Past-year food insecurity was measured through the Food Insecurity Experience Scale and categorised into 'none', 'low' and 'high'. Multinomial logistic regression was used to determine the crude and adjusted relative risk ratios (aRRRs) of IPV perpetration in relation to self-reported food insecurity, adjusting for sociodemographic and health risk behaviours.

**Results** The prevalence of self-reported lifetime IPV perpetration was 14.6% for physical and 6.5% for sexual violence, while 5.3% reported to have perpetrated both physical and sexual IPV. Most (75.7%) males reported no food insecurity, followed by low (20.7%) and high (3.6%) food insecurity. In adjusted models, food insecurity was associated with increased risk of having perpetrated both physical and sexual violence (aRRR=2.57, 95% CI 1.52 to 4.32). IPV perpetration was also independently associated with having had more than one lifetime sexual partner and drinking alcohol, but not with education level or religion.

**Conclusion** This study suggests that food insecurity is associated with male IPV perpetration, and more

## Strengths and limitations of this study

► This is the first study to investigate the association between lifetime male perpetration of intimate partner violence (IPV) with food insecurity in Uganda.
► We included a population-based, diverse and representative sample of participants from different geographical areas (rural, semiurban and urban) in Uganda.
► The cross-sectional nature of the study precludes any conclusion about the temporal association between IPV and food insecurity.
► Gender-based violence including IPV is a sensitive topic, and may be subject to under-reporting and social desirability bias. Nonetheless, this pre-pandemic assessment serves as an important baseline for further studies of the link between food insecurity and IPV during and after COVID-19-related lockdowns in Uganda.

efforts are needed to prevent and mitigate the expected worsening of this situation as a result of the COVID-19 pandemic.

## INTRODUCTION

Intimate partner violence (IPV) is a major global health problem and human rights violation. IPV consists of intentional and abusive attitudes in the form of physical, sexual and/or emotional offence and controlling behaviours within an intimate partnership.[1] While IPV can take different forms, male-perpetrated IPV is the most common form of violence against women (VAW).[2] Globally, one in three ever-partnered women reports experiencing some form of physical and/ or sexual violence in their lifetime, with the

WHO for the Africa region recording the highest prevalence (37%), followed by the Americas (29.8%).[3] The WHO multicountry study on VAW estimated that the prevalence of male IPV perpetration was 21.1% for physical violence only, 11.7% for sexual violence only, and 11.8% for both physical and sexual violence.[4] Reports from diverse social and cultural contexts have demonstrated that IPV perpetration begins early in the life course, with most perpetrators reporting that they first engage in it during adolescence.[5]

IPV carries multiple, well-established consequences for women's health and well-being, with femicide being the most extreme form.[6 7] IPV is associated with increased risk of sexually transmitted infections (STIs) including HIV,[8 9] severe depressive symptoms and substance use and abuse, such as binge drinking.[10] Furthermore, children born to women experiencing IPV are at higher risk of premature death, poor health outcomes, and emotional and behavioural problems later in life.[11] Experiencing IPV has been shown to affect women and girls' social and economic empowerment,[12] including decreased productivity at work, loss of employment opportunities and other important social engagement.[13]

Uganda has one of the highest burdens of IPV in the world and women's experiences of IPV have been the main focus of research on VAW in the country. The 2014 Ugandan Demographic and Health Survey (DHS) found that 47% of married women had experienced physical IPV in their lifetime, with 29% reporting lifetime sexual IPV.[14] Slapping, hitting and beating pregnant women were found to be common, and have been shown to worsen during the course of pregnancy in Northern Uganda.[15 16] Furthermore, the country's HIV incidence has been found to correlate with the frequency and duration of exposure to IPV.[17]

While multiple intersecting factors drive male perpetration of VAW, there is growing evidence linking IPV with food insecurity, most often related to poverty,[18 19] which is defined as 'a household-level economic and social condition of limited or uncertain access to adequate food'.[20] For example, a cross-sectional study in Kampala slums found that extreme physical violence perpetration and experience of such violence was common among a convenience sample of young men (14–24 years), and was found to correlate with reported hunger and alcohol consumption, illicit drug use, poor mental health status and parental neglect due to alcohol.[21] In a report from South Africa by Gibbs et al, economic indicators of food insecurity, such as unemployment and low earnings in the past month, were associated with IPV perpetration by men.[22] A small-scale study from Abidjan, Cote d'Ivoire revealed that women with severe forms of food insecurity were at higher risk of experiencing IPV.[23] While studies from other regions such as Nepal or the USA have established a significant relationship between food insecurity and both IPV perpetration and experiences among women, few studies from sub-Saharan Africa exist on this topic.[24 25]

According to the United Nations (UN), food insecurity is increasing in Uganda, rising from 24.1% in 2006 to 41% in 2018.[26] The ongoing COVID-19 crisis has contributed to this considerable increase in Ugandan food insecurity and, in line with predictions of the World Bank and the World Food Programme, is likely to continue to do so.[27 28] Likewise, there is great concern that IPV is increasing across the world as a consequence of social isolation, poverty and despair due to COVID-19-related lockdowns, social and financial hardships.[29]

Male IPV perpetration against women is a complex, multilevel, social, economic and structural problem, one rooted in unequal gender norms and systems of power. In most cultural settings, a gendered order exists that favours stereotypically masculine men's dominance over women as well as over other more marginalised masculinities that do not live up to the norms of being a 'real' man.[30] IPV can thus be viewed both as a symptom of and a tool to achieve this type of 'hegemonic' masculinity and power.[31] Being unable to provide food for one's self or household could be interpreted as failing to meet stereotypical masculine norms, thereby forcing men to use alternative strategies (such as violence) to demonstrate their manhood.[19] Mental health issues (depression or anxiety), which occur when concerns about food availability are accompanied by poor coping mechanisms such as alcohol consumption, can further lead to IPV as demonstrated by Hatcher et al in periurban South Africa.[19] Furthermore, the concept of 'patriarchal risk'[32] can help to theoretically explain the cultural and societal dependence of women on male family members for food and protection, and traditional gender roles are closely linked to all dimensions of food insecurity: access, availability, stability and utilisation. Although women tend to have less control over the household budget, they are often held responsible for feeding the family and blamed if they fail to provide food on the table, leading to different harmful coping strategies such as transactional sex, which in turn increases their risk of HIV.[33]

Given the increasing reports of food insecurity in Uganda[26] and the lack of existing literature in low-income and-middle income sub-Saharan African settings, it is important to investigate the potential link between food insecurity and male-perpetrated IPV in the country. The primary aims for the present study were to determine the prevalence of lifetime male IPV perpetration among a representative sample of males in two Ugandan districts, and to assess its association with food insecurity. The secondary aim was to determine whether the strength of the association between food insecurity and male IPV perpetration is affected by sociodemographic factors and health risk behaviours in the urban, semi-urban and rural populations under study in central Uganda. The evidence gained through this study can help inform programmes and policies to prevent and better respond to IPV in Uganda and beyond, something that likely will be needed more than ever as we enter into the post-pandemic era.

## METHODS

### Study design

Data were collected between May 2018 and July 2019 as part of a cross-sectional baseline survey of a longitudinal open population-level cohort established by the Africa Medical and Behavioural Sciences Organization (AMBSO) to conduct population health surveillance (PHS) in the Wakiso and Hoima districts of Uganda. In brief, the AMBSO PHS aims to generate evidence-based data in order to inform policy on the health status of the population through periodic monitoring of disease trends and determinants of health. The cohort includes males and females aged 13–80 years in the study communities and collects yearly data on sociodemographic factors, violence including IPV, non-communicable diseases (diabetes, hypertension, cancers and so on), communicable diseases (such as HIV and STIs), food and nutrition, immunisation of children, health risks (alcohol use, illicit drug consumption, number of sexual partners and male circumcision) and mental health. The current study uses the baseline data with a focus on men's lifetime perpetration of IPV in the study communities.

### Study communities

With a population of nearly 2 million, Wakiso district lies in the central region of Uganda and surrounds parts of the capital, Kampala. Hoima district, located in the Bunyoro Region of midwestern Uganda, has a smaller population of roughly 570 000 individuals. The selection of these two districts was purposeful, based on their common and diverse characteristics. Wakiso has a high migrant worker population and is known to be a hot spot for sex workers and men who have sex with men. The discovery of oil in Hoima and increasing mining activities attract migrant workers, sex workers and fisher folks—populations known to have high risk of HIV.[34] Each district was stratified into urban, semiurban and rural areas. Residents in all urban and semiurban areas, and those in a randomly selected rural community (representing the rural subcounties), were included in the baseline survey.

### Study population and sampling

The study population comprised of males aged 13–80 years in the study communities. A lower age limit of 13 years was considered because of the high prevalence of early sexual debut in Uganda.[35] This study sample was defined as males who reported ever having a sexual relationship. In each study community, the sampling frame included all households. In each household, all males aged 13–80 years were considered potential participants. Data from the Ugandan population census suggested that the total populations of male inhabitants in Wakiso and Hoima districts were 949 035 and 287 906, respectively.[36] In determining the sample size, the study population males aged 13–80 years was estimated to be 65% of the total male population in each district, that is, 621 617 and 188 578 in Wakiso and Hoima, respectively. The Ugandan DHS 40% estimate of male IPV perpetration[35] and the

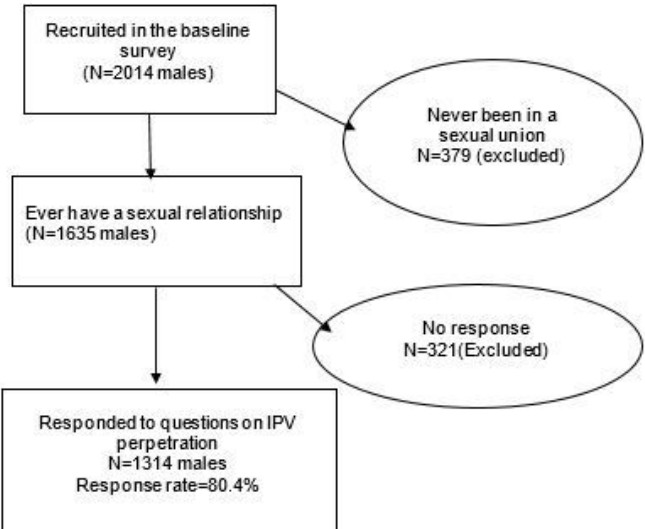

**Figure 1** Study flow, illustrating the selection process for inclusion into the analysis. IPV, intimate partner violence.

study population were entered into Epin Info (V.7.2.3.0) STATCalc tool to estimate the study sample. A minimum sample of N=738 was needed to estimate the association between food insecurity and IPV perpetration, with 80% power at the α=0.05 level. To be eligible, participants had to be able and willing to respond to the survey questions and to provide informed consent. The full survey sample consisted of N=2014 male participants who were recruited into the AMBSO PHS cohort. Males who reported not ever having been in a sexual union (N=379) and those who abstained from responding to questions on male IPV perpetration (N=321) were excluded (figure 1).

Even though the individual response rate was 80.4%, the final analysis sample (N=1314) was near double the estimated sample of N=738 participants needed to estimate the association. This 80.4% response rate is similar to that of a multicountry survey on male IPV perpetration in Asia.[4] A sensitivity analysis comparing the characteristics of those who responded to the IPV questions and those who did not respond is included in online supplemental appendix 1. The sensitivity analysis revealed that, in comparison with those who responded to the IPV questions, non-respondents were more likely to be young, Christian and of lower educational level. There was no significant difference in food insecurity among non-respondents and respondents. Any observed differences did not affect the findings as the final sample was more than double the minimum estimated sample needed to establish an association.

### Procedures

Data were collected in the baseline using a structured questionnaire covering 10 domains: sociodemographics; first sex experience; family planning practice; marital status and practices; food supply and food security; IPV; HIV testing services; HIV care and treatment services; prostate cancer and behaviour risk. Trained data collectors of the same sex as participants collected the data via

**Table 1** Definition and measurement of lifetime IPV perpetration (dependent variable)

| Type of violence perpetration | Have you ever done any of the following to your sexual partner? |
|---|---|
| Physical violence only ('yes' to ≥1 indicator) | ▶ Push, slap or hold partner down?<br>▶ Punch with fist or with something that could hurt partner?<br>▶ Kick or drag partner?<br>▶ Tried to strangle or burn partner?<br>▶ Attacked partner with a knife, gun or other types of weapon? |
| Sexual violence only ('yes' to ≥1 indicator) | ▶ Force her to perform sexual acts against partner's will?<br>▶ Used threats to force her to have sex when she did not want to?<br>▶ Physically force her to have sex against her wish? |
| Physical and sexual violence ('yes' to ≥1 indicators of both) | ▶ Perpetration of both physical violence and sexual violence |
| No IPV perpetration | ▶ No perpetration of both physical violence and sexual violence |

IPV, intimate partner violence.

face-to-face interviews, using tools translated into local languages (Luganda in Wakiso and Runyoro in Hoima).

In addition to these survey questions, the PHS collected 10 mL of venous blood samples that were used to determine the HIV and syphilis status of participants, and aliquots stored for other future studies. HIV status was determined following the Ugandan Ministry of Health rapid test algorithm,[37] and the rapid plasma regain serological test was used to determine syphilis status of the participants.

### Dependent variables
The dependent variable was adapted from the WHO multicountry study on VAW,[3] and defined as lifetime self-reported perpetration of physical and/or sexual violence against an intimate (sexual) partner. Following the approach used by Fulu *et al*,[4] the outcome was grouped into four independent and unordered categories: (1) no perpetration of physical and/or sexual violence (no IPV); (2) physical IPV perpetration only; (3) sexual IPV perpetration only; and (4) both physical and sexual IPV perpetration (table 1). The 'no IPV perpetration' category was used as the reference group, referring to males reporting to never have perpetrated physical and/or sexual violence against their intimate sexual partner. The 'physical violence only' category consisted of males reporting to have perpetrated one or more forms of physical violence, but who had never used any sexual violence; 'sexual violence only' was coded using the same approach but referring to one or more forms of sexual violence. Males who reported to having perpetrated both physical and sexual violence in their lifetime were grouped into

the 'physical and sexual violence' category. Lifetime estimates were used in this study in order to capture all self-reports of IPV perpetration.

### Independent variables
#### Food insecurity
The main independent variable was past-year food insecurity, assessed using an adapted version of the Food Insecurity Experience Scale questions[38] in which six selected items were used to evaluate the household food situation during the preceding 12 months (table 2).

In line with international definitions,[39] negative responses to assessment questions such as 'no' 'never' 'only 1 or 2 months' were coded as '0' (referring to those who reported having minor access problems for food during the preceding 12 months), while positive or affirmative responses such as 'yes', 'always', 'almost every month', 'some months but not every month' were coded as '1' (referring to having problems with access to adequate food during the preceding 12 months). All codes were summed to obtain a total score of up to 6 (ie, 0–6) for food insecurity, with higher scores indicating the extent of food insecurity. As adapted from international approaches,[39] participants were categorised into the following categories: no food insecurity (0–1), low food insecurity (2–4) and high food insecurity (hunger) (5–6). Due to the small proportion reporting high food insecurity (3.6%), low food insecurity and high food insecurity were merged during the modelling procedures in line with previous studies,[19] that is, comparing food insecurity against food security in line with previous studies.

#### Sociodemographic and health risk behaviours
Other important independent variables included sociodemographic indicators such as: study district (urban/semiurban/rural); age; education level (categorised into primary school and below, secondary school and post-secondary); marital status (married or in a union/not married); religion (categorised into Christian, non-Christians); and having living children (yes/no). Health risk behaviours included: number of lifetime sexual partners (one partner/two or more partners); past-year condom use (yes/no); use of illicit drugs (yes/no); alcohol consumption (yes/no) and male circumcision (yes/no). The choice of health risk behaviours was based on previous studies indicating number of sexual partners as well as (low) condom use[40–42] and male circumcision[42] as risk factors for IPV.

### Statistical analysis
We first conducted descriptive analyses to explore the distribution of variables and to identify outliers. The analysis in this report was not weighted. Missing values for covariates were replaced using multiple imputation assuming random missingness. Pearson's $X^2$ test and Fisher's exact test were used to compare proportions with α set at 5%. Next, we performed bivariate multinomial logistic regression to determine the crude relative risk

**Table 2** Assessment questions and coding for the measurement and classification of food insecurity among participants in the study

| Question | Responses | Coding of responses |
|---|---|---|
| Q.1 In the past 12 months, were there months in which you did not have enough food to meet your family's needs? | Yes/no | Yes=1, no=0 |
| Q.2 In the last 12 months, have you or other adults in your household withheld a meal because there was not enough food? | Yes/no | Yes=1, no=0 |
| Q.3 In the last 12 months, did you or other adults in your household ever not eat for a whole day because there wasn't enough food? | Yes/no | Yes=1, no=0 |
| Q.4 In the last 12 months, did you ever cut the size of any of the children's meals because there wasn't enough food? | Yes/no | Yes=1, no=0 |
| Q.5 In the last 12 months, how often were you worried that food would run out? | Always/sometimes/never | Always=1, sometimes=1, never=0 |
| Q.6 If yes in Q.5, how often did this happen? | Almost every month/ sometimes but not every month/only 1 or 2 months/ don't remember | Almost every month=1, sometimes but not every month=1, only 1 or 2 months=0, don't remember=0 |

Source: Food Insecurity Experience Scale.

ratios (RRRs) and 95% CIs of IPV perpetration in relation to the independent variables. We followed the approach used by Fulu et al[4] and used 'No IPV perpetration' as the base outcome against which all other outcomes were compared, assuming that the association of food insecurity is different for each typology of violence. This type of regression produces RRRs, which are the exponentiated coefficients of the regression, and explain the relative effect of the independent variable on the outcomes using one of the outcomes as a base outcome. A full model was built to explain the associated factors to IPV perpetration in relation to both food insecurity, health risk behaviours and sociodemographic background. The full model (adjusting for food insecurity and sociodemographic and health risk behaviours) included all the variables from the bivariate analysis, irrespective of significance levels, that are theoretically known to be associated with IPV perpetration. Variables for the final adjusted model were selected through backward elimination using a maximum likelihood ratio test. All analyses were conducted using STATA V.16 (StataCorp), while Microsoft Office Excel was used to design figures.

### Patient and public involvement

The study team worked hand in hand with community health workers, local community leaders and a community-research advisory structure to develop and clarify a common understanding of the aims of the population-based survey, the design and operational aspects as well as how results will be disseminated. As part of this 'patient and public involvement' effort, a consensus was reached with the advisory committee on a plan to increase general public awareness of the study and health issues that were deemed areas of PHS focus.

## RESULTS

### Sample characteristics

Table 3 shows the characteristics of the study participants. The mean age of respondents was 34 years (SD: ±12.9). Food insecurity was found to affect one in four men, with 20.7% and 3.6% categorised as low versus high food insecurity, respectively. About one-third and 50% of men reported having had multiple sex partners and used alcohol in the past year, respectively; and approximately 6% tested positive for HIV.

### Prevalence of lifetime IPV perpetration

Table 4 presents the prevalence of perpetration of different forms of IPV against an intimate (sexual) partner by male participants. Most (73.6%) males reported never having perpetrated any form of IPV. The prevalence of self-reported physical IPV perpetration ranged from 19.3% (pushing, slapping, holding down) to 0.9% (attacked partner with a knife, gun or other weapon), with 14.6% having perpetrated any form of physical violence only. Lifetime perpetration of sexual IPV ranged from 8.2% (threatened or pressured partner into sex when unwanted) to 5.4% (physically forced partner to have sex) and 2.4% (used other types of force to have sex), with 6.5% of males reporting having perpetrated any form of sexual IPV only. In total, 5.3% of males reported previous perpetration of both physical and sexual IPV.

**Table 3** Sample characteristics of male respondents in the two study districts in Uganda (N=1314)

| Variable | n | % |
|---|---|---|
| Age (mean, SD) | 34±12.9 | |
| Education | | |
| Primary and below | 688 | 52.4 |
| Secondary school | 477 | 36.3 |
| Post-secondary | 149 | 11.3 |
| Marital status | | |
| Married or in a union | 1163 | 88.5 |
| Not married | 151 | 11.5 |
| Have any living children | | |
| Yes | 999 | 76.2 |
| No | 312 | 23.8 |
| Religion | | |
| Christianity | 1068 | 81.3 |
| Non-Christian | 246 | 18.7 |
| Past-year food security | | |
| No food insecurity | 995 | 75.7 |
| Low food insecurity | 272 | 20.7 |
| High food insecurity | 47 | 3.6 |
| Score (mean, SD) | 1.48±1.05 | |
| Number of lifetime sexual partners | | |
| One partner | 919 | 69.9 |
| Two or more partners | 395 | 30.1 |
| Current use of condoms | | |
| Yes | 519 | 39.5 |
| No | 795 | 60.5 |
| HIV status from blood sample | | |
| Negative | 1224 | 94.3 |
| Positive | 74 | 5.7 |
| Past-year alcohol use | | |
| Yes | 677 | 48.5 |
| No | 640 | 51.5 |
| Circumcised | | |
| Yes | 659 | 50.2 |
| No | 655 | 49.8 |
| Past-year use of illicit drugs | | |
| Yes | 43 | 3.3 |
| No | 1271 | 96.7 |

Figure 2 shows the prevalence of self-reported IPV perpetration across different age groups. As can be seen, the prevalence of physical violence only and both physical and sexual violence was highest among those aged 25–34 years old compared with the other age groups (p<0.001), with perpetration of sexual violence only being most commonly reported by young men (18–24 years) and less common after age 34 years (p=0.517).

**Table 4** Prevalence of self-reported lifetime perpetration of IPV against a sexual partner among ever-partnered males aged 13–80 years in two districts of Uganda, May 2018–July 2019

| | Lifetime IPV perpetration (N=1314) | |
|---|---|---|
| | n | % |
| Physical violence (any form) | 261 | 19.9 |
| Push, slapped or held down partner | 253 | 19.3 |
| Kicked or dragged partner | 54 | 4.1 |
| Tried to strangle partner | 19 | 1.5 |
| Attacked partner with a knife, gun or other weapon | 12 | 0.9 |
| Punch with fist or with something that could hurt partner | 61 | 4.6 |
| Sexual violence (any form) | 155 | 11.8 |
| Threatened or pressured partner into sex when unwanted | 108 | 8.2 |
| Physically forced partner into sex | 70 | 5.4 |
| Used other ways to force partner to perform sexual acts when unwanted | 31 | 2.4 |
| Any physical or sexual violence | 347 | 26.4 |
| Physical IPV perpetration only | 192 | 14.6 |
| Sexual IPV perpetration only | 86 | 6.5 |
| Both physical and sexual IPV perpetration | 69 | 5.3 |
| No IPV perpetration | 967 | 73.6 |

IPV, intimate partner violence.

### Factors associated with IPV perpetration
#### Bivariate analysis
In bivariate analysis, food-insecure participants had an RRR of 2.71 (95% CI 1.64 to 4.46) to have perpetrated both physical and sexual violence, but no significantly higher risk to have perpetrated physical versus sexual violence only, respectively (table 5). Compared with the youngest age group 13–24 years, young adult participants (25–34 years) had an RRR of 2.59 (95% CI 1.64 to 4.07) for perpetrating physical violence only, and 2.64 (95% CI 1.30 to 5.36) for both physical and sexual violence, and the RRR for those aged 35+ years was 2.16 (95% CI 1.38 to 3.38) for physical violence. Being married was associated with a lower relative risk of having perpetrated sexual violence only. Having living children and reporting no condom use were associated with perpetration of physical violence only. Reported use of illicit drugs during the preceding year was associated with perpetration of physical violence only and both physical and sexual violence. There was no association between education level with IPV perpetration.

#### Multivariable analysis
Table 6 illustrates the adjusted RRR (aRRR) for IPV perpetration in relation to food insecurity, controlling

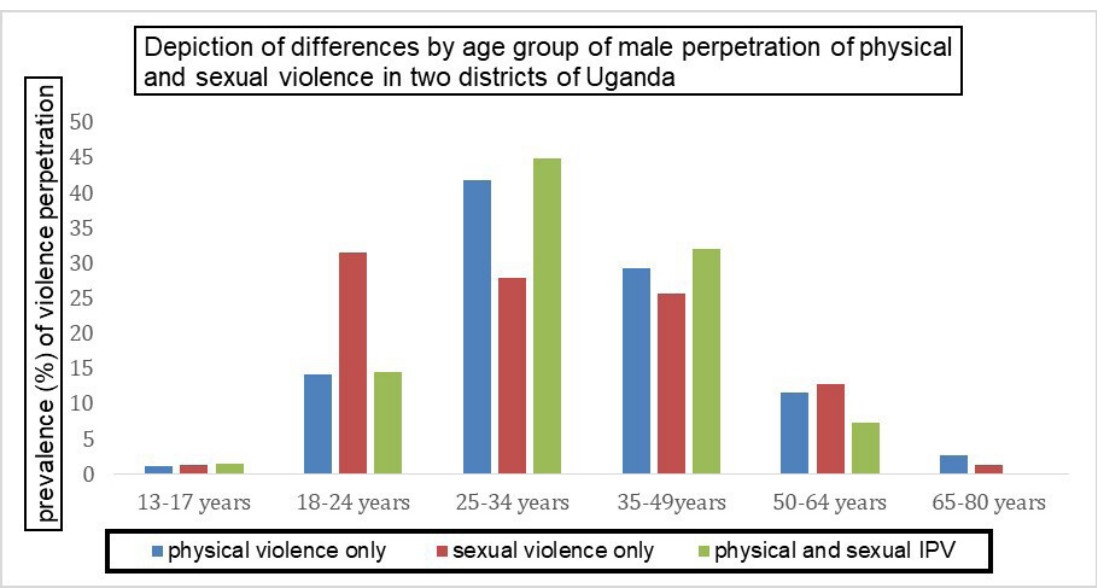

**Figure 2** Depiction of lifetime prevalence by age group of male lifetime IPV perpetration in two districts of Uganda, May 2018–July 2019. IPV, intimate partner violence.

for sociodemographic factors and health risk behaviours. Food insecurity remained significantly associated with self-reported perpetration of both physical and sexual violence (aRRR=2.57, 95% CI 1.52 to 4.32). In terms of independent health risk factors, having two or more lifetime sexual partners also remained associated with perpetration of physical violence only, and both physical and sexual violence. Any alcohol consumption remained associated with all typologies of IPV perpetration under study, and illicit drug use remained associated with physical violence and both physical and sexual violence. In terms of sociodemographics, having living children remained associated with perpetration of physical violence only, while being married appeared to be protective against perpetration of sexual violence only. Age and religion were not associated with any IPV perpetration in the adjusted model.

## DISCUSSION

The current study set out to explore the prevalence of lifetime male IPV perpetration and its association with food insecurity as well as sociodemographics and health risk behaviours in two Ugandan districts. We found that one in four men reported ever perpetrating any form of physical and/or sexual violence against their intimate partners, and 5.3% reported perpetrating both physical and sexual IPV. Past-year food insecurity was associated with male perpetration of both physical and sexual violence after adjusting for age group, religion, number of lifetime sexual partners, marital status, having living children, alcohol use and illicit drug use.

The prevalence of lifetime perpetration of IPV reported by this study is lower than that of other studies conducted in the region.[19 43 44] Obtaining accurate data on the true burden of VAW is a global concern due to

under-reporting.[45] It is likely that community awareness about laws and litigations in place to prevent VAW may undermine truthful responses to IPV survey questions. Nonetheless, a fair proportion of men in our survey did admit to IPV perpetration. In these new demographic surveillance sites and first round of the assessments, it is possible that the communities have not yet built sufficient trust with the survey teams to report very sensitive information and potential illegal behaviours. Comparing the prevalence estimates of this study with data from previous studies is therefore difficult, mainly due to the existence of limited research in a sub-Saharan African context, and the non-standardised ways in which surveys have assessed male IPV perpetration. For instance, although the 2006 Ugandan DHS[14] found that 40% of males perpetrated IPV, this finding was solely based on reports of physical violence (eg, partner hitting, slapping and kicking) without taking into account sexual IPV perpetration. Another large-scale study that focused on past-year perpetration (rather than lifetime) in the Rakai district of Uganda indicated a prevalence of 10.4% for physical violence and 3.1% for sexual violence.[42] In the present study, 19.3% of men reported ever slapping or pushing their intimate partners, an estimate that is similar to other countries including Ghana (17.4%)[43] and Vietnam (23.5%),[46] but lower than those obtained by the International Men and Gender Equality Study in the Democratic Republic of Congo (45.1%) and Rwanda (39.1%).[44]

In this pre-pandemic assessment, one in four men reported to be food insecure, and food insecurity was associated with almost three times the relative risk of having perpetrated both physical and sexual IPV. Our findings are consistent with results of a recent study from a periurban setting of South Africa, in which food insecurity doubled the odds of males' perpetration of IPV.[19] The

**Table 5** Crude relative risk ratios (RRRs) of factors associated with male IPV perpetration in two districts of Uganda (baseline outcome=noIPV) (N=1314), 2019

| Variable | N | No IPV N (%)* | Physical violence only N (%)* | RRR (95% CI) | P value | Sexual violence only N (%)* | RRR (95% CI) | P value | Both physical and sexual violence N (%)* | RRR (95% CI) | P value |
|---|---|---|---|---|---|---|---|---|---|---|---|
| **Age group (years)** | | | | | | | | | | | |
| 13–24 (ref) | 353 | 285 (80.7) | 29 (8.2) | | | 28 (7.9) | | | 11 (3.1) | | |
| 25–34 | 439 | 304 (69.3) | 80 (18.2) | 2.59 (1.64 to 4.07) | 0.001 | 24 (5.5) | 0.80 (0.46 to 1.42) | 0.451 | 31 (7.1) | 2.64 (1.30 to 5.36) | 0.007 |
| 35+ | 522 | 378 (72.4) | 83 (15.9) | 2.16 (1.38 to 3.38) | 0.001 | 34 (6.5) | 0.92 (0.54 to 1.54) | 0.741 | 27 (5.2) | 1.85 (0.90 to 3.79) | 0.093 |
| **Education** | | | | | | | | | | | |
| Primary or below (ref) | 688 | 499 (72.5) | 106 (15.4) | | | 49 (6.7) | | | 37 (5.4) | | |
| Secondary school | 477 | 349 (73.2) | 68 (14.3) | 0.92 (0.66 to 1.28) | 0.612 | 34 (7.1) | 1.05 (0.66 to 1.68) | 0.815 | 26 (5.5) | 1.0 (0.60 to 1.69) | 0.986 |
| Post-secondary | 149 | 119 (79.9) | 18 (12.1) | 0.71 (0.42 to 1.21) | 0.216 | 6 (4.0) | 0.55 (0.23 to 1.31) | 0.176 | 6 (4.0) | 0.68 (0.28 to 1.65) | 0.393 |
| **Religion** | | | | | | | | | | | |
| Islam and others (ref) | 242 | 186 (76.9) | 24 (9.9) | | | 16 (6.6) | | | 16 (6.6) | | |
| Christian religion | 1072 | 781 (72.8) | 168 (15.7) | 1.66 (1.06 to 2.63) | 0.028 | 70 (6.5) | 1.04 (0.59 to 1.83) | 0.887 | 53 (4.9) | 0.79 (0.44 to 1.41) | 0.424 |
| **Marital status** | | | | | | | | | | | |
| Not married (ref) | 151 | 109 (72.2) | 15 (9.9) | | | 18 (11.9) | | | 9 (5.9) | | |
| Married | 1163 | 858 (73.8) | 177 (15.2) | 1.50 (0.85 to 2.63) | 0.159 | 68 (5.9) | 0.48 (0.27 to 0.84) | 0.010 | 60 (5.2) | 0.85 (0.41 to 1.75) | 0.655 |
| **Have living children** | | | | | | | | | | | |
| No (ref) | 312 | 258 (82.7) | 18 (5.7) | | | 25 (8.0) | | | 11 (3.5) | | |
| Yes | 999 | 708 (70.9) | 172 (17.2) | 3.48 (2.10 to 5.77) | <0.001 | 61 (6.1) | 0.89 (0.54 to 1.45) | 0.636 | 58 (5.8) | 1.92 (0.99 to 3.71) | 0.053 |
| **Past-year alcohol consumption** | | | | | | | | | | | |
| No (ref) | 637 | 511 (80.2) | 74 (11.6) | | | 32 (5.0) | | | 20 (3.1) | | |
| Yes | 677 | 456 (67.4) | 118 (17.4) | 1.79 (1.30 to 2.45) | <0.001 | 54 (7.9) | 1.89 (1.20 to 1.98) | 0.006 | 49 (7.3) | 2.74 (1.61 to 4.68) | <0.001 |
| **Past-year illicit drug** | | | | | | | | | | | |
| No (ref) | 1271 | 948 (74.6) | 180 (14.2) | | | 83 (6.5) | | | 60 (4.7) | | |
| Yes | 43 | 19 (44.2) | 12 (27.9) | 3.32 (1.59 to 6.97) | 0.001 | 3 (6.9) | 1.80 (0.52 to 1.22) | 0.351 | 9 (20.9) | 7.48 (3.24 to 17.2) | <0.001 |
| **Food insecure** | | | | | | | | | | | |
| No food insecurity (ref) | 995 | 756 (76.0) | 139 (14.1) | | | 56 (5.7) | | | 31 (3.2) | | |
| Food insecure | 319 | 214 (67.1) | 52 (16.3) | 1.31 (0.92 to 1.86) | 0.395 | 23 (7.2) | 1.28 (0.77 to 1.22) | 0.097 | 30 (9.4) | 2.71 (1.64 to 4.46) | <0.001 |
| **Condom use in past 12 months** | | | | | | | | | | | |
| No (ref) | 795 | 571 (71.8) | 130 (16.4) | | | 50 (6.3) | | | 44 (5.5) | | |
| Yes | 519 | 396 (76.3) | 62 (11.9) | 0.69 (0.49 to 0.96) | 0.026 | 36 (6.9) | 1.04 (0.66 to 1.62) | 0.870 | 25 (4.8) | 0.82 (0.49 to 1.36) | 0.441 |
| **Circumcised** | | | | | | | | | | | |
| No (ref) | 655 | 464 (70.8) | 107 (16.3) | | | 43 (6.6) | | | 41 (6.3) | | |
| Yes | 659 | 503 (76.3) | 85 (12.9) | 0.73 (0.54 to 1.00) | 0.050 | 43 (6.5) | 0.92 (0.59 to 1.43) | 0.720 | 28 (4.3) | 0.63 (0.38 to 1.03) | 0.068 |

**Table 5** Continued

| Variable | N | No IPV N (%)* | Physical violence only | | | Sexual violence only | | | Both physical and sexual violence | | |
|---|---|---|---|---|---|---|---|---|---|---|---|
| | | | N (%)* | RRR (95% CI) | P value | N (%)* | RRR (95% CI) | P value | N (%)* | RRR (95% CI) | P value |
| Number of lifetime sexual partner | | | | | | | | | | | |
| One (ref) | 919 | 791 (76.3) | 124 (13.5) | | | 55 (6.0) | | | 39 (4.2) | | |
| Two or more | 395 | 266 (67.3) | 68 (17.2) | 1.44 (1.04 to 2.01) | 0.028 | 31 (7.8) | 1.48 (0.94 to 2.36) | 0.098 | 30 (7.5) | 2.03 (1.23 to 3.33) | 0.005 |
| Hoima district (ref: Wakiso) | 736 | 526 (71.5) | 112 (15.2) | 1.17 (0.86 to 1.61) | 0.912 | 55 (7.5) | 1.49 (0.94 to 2.35) | 0.679 | 43 (5.8) | 1.38 (0.84 to 2.29) | 0.875 |

*Row percentage.
IPV, intimate partner violence.

results from our study align with findings from Nepal[25] and the USA[24] that food insecurity was more common among married women who had experienced physical violence from their intimate sexual partners; however, the causal relationships are uncertain. The World Food Programme expects the number of people who are food insecure to increase dramatically as a consequence of COVID-19-related mobility restrictions and poverty, potentially driving another 135 million people worldwide onto the brink of starvation. Similarly, the UN Population Fund warns of increases in IPV as a consequence of social isolation, hopelessness and financial hardship.[29] Thus, the pre-pandemic estimates presented in this study are expected to worsen and should serve as an important warning indicator for urgent mitigation efforts.

Previous research indicates that the association between food insecurity and IPV perpetration may be driven by mental health problems such as anxiety and depression that arise from concerns about food availability related to poverty and unemployment,[19] which are also expected to increase as a result of the current pandemic. The influence of conservative gender norms is important, whereby the inability of men to provide for their partners and households affects their perceived masculinity and contributes to their use of violence.[47] A qualitative study in Bangladesh found that inadequate food portions offered to men may trigger retaliatory acts of violence against their wives, and that men could withhold resources used to acquire food as a form of power.[48] Corroborated by previous research, the present study also found that men who had consumed alcohol in the past year and who have multiple lifetime sexual partners were at significantly higher risk of perpetrating IPV.[4 49 50] The use of illicit drugs, though assumed to be under-reported, was also found to be associated with IPV perpetration after adjusting for confounding, in line with previous research.[51] The association between food insecurity and IPV perpetration may thus be influenced and mediated by multiple factors previously examined, for example, through structural equation models including poor mental health, gender attitudes, multiple partnerships, controlling behaviours and alcohol consumption.[19 22]

Furthermore, we found that the prevalence of IPV perpetration varied by age group, with reports of sexual and physical IPV being more common among young adult males compared with older men, while in the adjusted models, being 35 years and older appears to confer a protective effect against perpetraton of IPV. These findings are in line with previous studies from the USA.[50 52] For young men, it is possible that a relative lack of relationship experiences, masculine-identity seeking and vulnerability to peer influence[50 52] increase their tendency to perpetrate violence.[53] These findings have implications for the design of future preventive interventions for different subpopulations, especially men in the younger age groups.

Our findings highlight the need for early prevention of IPV that targets young men, and the necessity

**Table 6** Adjusted relative risk ratios (aRRRs) of factors associated with male IPV perpetration in two districts of Uganda (N=1314)

| Variable* | Physical violence only† | | | | Sexual violence only† | | | | Both physical and sexual violence† | | | |
|---|---|---|---|---|---|---|---|---|---|---|---|---|
| | Crude RRR (95% CI) | P value | aRRR (95% CI) | P value | Crude RRR (95% CI) | P value | aRRR (95% CI) | P value | Crude RRR (95% CI) | P value | aRRR (95% CI) | P value |
| Food insecurity (ref: no food insecurity) | 1.31 (0.92 to 1.86) | 0.395 | 1.19 (0.82 to 1.73) | 0.386 | 1.28 (0.77 to 2.12) | 0.235 | 1.33 (0.54 to 1.55) | 0.097 | 2.71 (1.64 to 4.46) | <0.001 | 2.57 (1.52 to 4.32) | 0.002 |
| Health risk behaviours | | | | | | | | | | | | |
| Two or more sexual partners (ref: one partner) | 1.0 (1.04 to 2.01) | 0.028 | 1.64 (1.15 to 2.33) | 0.012 | 1.48 (0.94 to 2.36) | 0.243 | 1.38 (0.85 to 2.25) | 0.098 | 2.03 (1.23 to 3.33) | 0.005 | 1.97 (1.15 to 3.37) | 0.016 |
| Drink alcohol (ref: no) | 1.79 (1.30 to 2.45) | <0.001 | 1.51 (1.07 to 2.12) | 0.006 | 1.89 (1.20 to 2.98) | 0.002 | 1.89 (1.16 to 3.07) | <0.001 | 2.74 (1.61 to 4.68) | <0.001 | 2.55 (1.42 to 4.58) | <0.001 |
| Illicit drug use the past year (ref: no) | 3.32 (1.59 to 6.97) | 0.003 | 3.84 (1.73 to 8.52) | 0.001 | 1.80 (0.52 to 6.22) | 0.657 | 1.29 (0.36 to 4.58) | 0.351 | 7.48 (3.24 to 17.2) | <0.001 | 4.80 (1.93 to 11.9) | 0.005 |
| Sociodemographic factors | | | | | | | | | | | | |
| 25–34 years (ref: 18–24 years) | 2.59 (1.64 to 4.07) | 0.001 | 1.37 (0.99 to 1.91) | 0.069 | 0.80 (0.46 to 1.42) | 0.132 | 0.84 (0.51 to 1.39) | 0.451 | 2.64 (1.30 to 5.36) | 0.007 | 1.66 (0.99 to 2.78) | 0.079 |
| 35+ years | 2.16 (1.38 to 3.38) | 0.001 | 0.95 (0.53 to 1.68) | 0.346 | 0.92 (0.59 to 1.83) | 0.325 | 0.80 (0.41 to 1.54) | 0.741 | 0.79 (0.44 to 1.41) | 0.093 | 0.40 (0.19 to 0.84) | 0.025 |
| Christian religion (ref: non-Christians) | 1.66 (1.06 to 2.63) | 0.028 | 1.53 (0.94 to 2.51) | 0.127 | 1.04 (0.59 to 1.83) | 0.203 | 0.82 (0.45 to 1.51) | 0.887 | 0.79 (0.44 to 1.41) | 0.424 | 0.61 (0.32 to 1.17) | 0.156 |
| Married (ref: not married) | 1.50 (0.85 to 2.63) | 0.159 | 1.34 (0.76 to 2.44) | 0.321 | 0.48 (0.27 to 0.84) | 0.021 | 0.49 (0.27 to 0.86) | 0.010 | 0.85 (0.41 to 1.75) | 0.655 | 0.81 (0.37 to 1.75) | 0.217 |
| Have children (ref: no) | 3.48 (2.10 to 5.77) | <0.001 | 3.24 (1.91 to 5.52) | 0.011 | 0.89 (0.54 to 1.45) | 0.325 | 0.92 (0.55 to 1.56) | 0.636 | 1.92 (0.99 to 3.71) | 0.053 | 1.77 (0.87 to 3.61) | 0.876 |

*Food insecurity was adjusted for age group, marital status, number of sexual partners, illicit drug use, past-year alcohol consumption, having living children and religion.
†Reference/base outcome: no IPV perpetration.
IPV, intimate partner violence.

to investigate and respond to food insecurity within the framework of IPV prevention. Even though a temporal association between IPV and food insecurity is yet to be established, the findings of this study should serve as an alert to more initiatives to address IPV, particularly in light of an expected dramatic increase in both IPV and food insecurity reports as a consequence of the COVID-19 pandemic.[29] Existing IPV preventive interventions in Uganda such as the SASA community-based programme[54] and the Program P initiative[55] may provide concrete guidelines for efforts to engage more men in IPV preventive programmes. The SASA and Program P interventions also use comprehensive strategies (including HIV risk reduction) and should be strengthened with considerations for advocacy to include for food security measures as part of the packages to respond to IPV. There is an urgent need to adapt context-relevant interventions to address food insecurity, especially during the current pandemic with severe increases in both IPV and food insecurity warning and reports worldwide. This is of particular importance in sub-Saharan Africa, where a huge surge in poverty rates and food insecurity is expected.[56 57] Thus, our study suggests that IPV may become an even larger public health issue in Uganda and beyond in the near future. Potential interventions to reduce poverty and food insecurity could include: efforts to boost agricultural food production such as short-term provision of seeds and tools to rehabilitate farming and facilitate the acquisition of credits for small businesses; conditional cash transfers to women household heads to enable purchase of necessities and pay for health expenses; implementation of school fee waivers and school feeding programmes; and the provision of food vouchers to enable purchase of food in local markets and the emergency distribution of food.[58] Structural interventions to address IPV should be advocated for, including more discussion on gender norms and harmful masculinities in schools and communities, putting in place systems to prevent male violence perpetration, encouraging women to report incidences of IPV accompanied with a strengthening of legislation, training and monitoring of the police force and justice departments in IPV-related issues and set up support and counselling centres for women who experience IPV.[59]

## LIMITATIONS

This study is one of the first to investigate lifetime perpetration of different forms of physical and sexual violence among both younger and older men, and to examine their association with food insecurity and male IPV perpetration in a population-based, representative cohort in rural, urban and semiurban Uganda. Given the self-reported nature of the data on a sensitive topic like IPV, underreporting of physical and sexual IPV as well as recall bias is possible. Some participants may have considered the questions to be too private and/or shameful to talk about, and some may have feared legal implications (given that IPV is illegal), or found it challenging to respond honestly.

In light of this potential social desirability bias, the true rate of IPV perpetration is likely higher than reported in this first round of the assessments. In the future, exposure to and increased population trust in the data collectors in this reoccurring population-based survey may further improve the confidence in self-reporting IPV perpetration. Even though the minimum sample needed for this study was attained, a fairly high proportion of men did not respond to the IPV questions in the survey. The sensitivity analysis revealed, however, that this had no significant effect on the findings. Data on poverty indicators such as income levels and employment status were unfortunately lacking at individual level, but it can be assumed that food insecurity is a reflection of income level and the availability and access to livelihood resources. Finally, due to the cross-sectional nature of the data coupled with the measurement of food insecurity as past year and IPV as lifetime, we were unable to establish a temporal association between food insecurity and IPV perpetration. Even though multivariate analysis was performed in this study, the likelihood of residual confounding still remains.

## CONCLUSIONS

Although likely an underestimation, the reported prevalence of male IPV perpetration in these Ugandan communities was still sizeable—in particular among young men, and must be addressed by enhanced IPV preventive strategies and programming in schools and communities. Food insecurity was associated with both physical and sexual IPV, suggesting the need for integrated approaches to address VAW, poverty and food insecurity. Considering the impact of COVID-19 lockdowns and multiple reports on increased IPV, as well as expected huge increases in poverty rates and food insecurity throughout Uganda, the need for preventive interventions is even greater than before. Longitudinal surveys are needed to investigate the temporal relationships and drivers of food insecurity in Uganda and beyond, and to determine its impact on different forms of violence.

**Acknowledgements** We thank the staff of Africa Medical and Behavioral Sciences Organization (AMBSO); the PHS study participants; the local community leadership; and the Wakiso and Hoima Districts Directorates of Health services for supporting this work.

**Contributors** The original ideas leading to the population health surveillance study, from which this manuscript is derived, were put together by SM, FN, GK, GN, SW and AME. GA and AK conceptualised the research questions for the current study. GA analysed the data, wrote the first draft of the study and oversaw the manuscript development process. SM, FN, SW and GK coordinated the participant recruitment as well as the data collection, management and entry. RBM, LZ, GN, GKN, EK and JN participated in actual data collection from the study participants. CFS and SR contributed to describing the measurement and classification of food insecurity. AK provided oversight to the study, providing edits, structure and scope to the study together with AME. All authors reviewed the drafts leading to this final manuscript.

**Funding** This study was made possible by private funds from AMBSO directors, the Karolinska Institute through a Swedish Research Council grant (grant number: 2019-04474), and the Swedish Institute which supported the first author with an educational study grant (ref: 02205/2019).

**Competing interests** None declared.

**Patient consent for publication** Not required.

**Ethics approval** The AMBSO PHS cohort study received ethical approval from the Clarke International University Ethics Committee (ref: CIUREC/0059) as the local Institutional Review Board (IRB) of record, and clearance from the Ugandan Council of Science and Technology (ref: SS4468). Clearance was obtained from both districts to conduct the study. All participants provided a written informed consent/ assent before participating in the study. With the exception of emancipated minors (under 18 years) who (like adults) gave their own written or oral assent in addition to written parental/guardian consent was obtained. Data were collected by a study team trained and certified in human subject protection and Good Clinical Practice. Interviews were conducted in private. Unique identification codes were used in all computerised data, and the lists that linked participants' names to the codes were accessible only to senior project staffs in password-protected computers. Participants who tested positive for HIV infection and syphilis were linked to specialised care while those who needed other health support services were referred to health facilities of their choice.

**Provenance and peer review** Not commissioned; externally peer reviewed.

**Data availability statement** Data are available upon reasonable request. The data for this study are strictly not available to the public. A reasonable request for the data can be done by contacting SM (smugamba@gmail.com).

**Open access** This is an open access article distributed in accordance with the Creative Commons Attribution 4.0 Unported (CC BY 4.0) license, which permits others to copy, redistribute, remix, transform and build upon this work for any purpose, provided the original work is properly cited, a link to the licence is given, and indication of whether changes were made. See: https://creativecommons.org/licenses/by/4.0/.

**ORCID iDs**
George Awungafac http://orcid.org/0000-0002-5859-0336
Carl Fredrik Sjöland http://orcid.org/0000-0003-3594-8849

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
