## [Reviewer comments · BMJ Open]

ARTICLE DETAILS

TITLE (PROVISIONAL)	Household food insecurity and its association with self-reported male perpetration of intimate partner violence: a survey of two districts in Central and Western Uganda
AUTHORS	Awungafac, George; Mugamba, Stephen; Nalugoda, Fred; Sjöland, Carl Fredrik; Kigozi, Godfrey; Rautiainen, S; Malyabe, Robert; Ziegel, Leo; Nakigozi, Gertrude; Nalwoga, Grace; Kyasanku, Emmanuel; Nkale, James; Watya, Stephen; Ekström, Anna; Kågesten, Anna

VERSION 1 – REVIEW

REVIEWER	Sheila Harvey London School of Hygiene & Tropical Medicine, UK
REVIEW RETURNED	19-Nov-2020

GENERAL COMMENTS	Thank you for the invitation to review this manuscript. The study addresses the need for more data on male perpetration of violence against women. General comment The manuscript needs a thorough proof-read and copy-edit as there are a number of typographical errors, e.g. should be temporal not temporary (3rd bullet in the strengths and limitations section). Abstract The authors state that the aim of the study is to determine lifetime prevalence of male perpetrated IPV, and to assess its association with food insecurity, but it seems that the investigators were looking at a range of factors, such as age, education etc. This needs to be made clear. In the conclusion the authors state that food insecurity could worsen with the SARS-CoV-2 outbreak – I would suggest adding to the introduction section that the prevalence of food insecurity has increased considerably in Uganda, which is likely to worsen with the SARS-CoV-2 outbreak. In the results, the authors state that 25-34 year-olds reported the highest prevalence of both physical and sexual violence – it would be helpful to include the prevalence for each form of violence. In the methods the authors state that food insecurity was categorised as no, low and high but there is no reference to these categories in the results section.
--

	The authors state in the results that there was no association between IPV perpetration and age but in the conclusion state that perpetration varied by age. Introduction The authors state that there are no previous studies examining the association between food insecurity and male perpetrated IPV in sub-Saharan settings, however, there are some studies such as Gibbs et al (2018) that have examined associations between poverty (including food insecurity) and IPV (including male perpetration). It would be better to state that there are few studies/limited research. The aim needs to be stated more clearly. In addition to measuring prevalence of male perpetrated IPV, the authors investigated the association of a range of factors with IPV, not just food insecurity, as stated. Methods The description of this study/cross-sectional analysis in relation the AMBSO cohort study is slightly confusing. The authors state the AMBSO cohort study is aimed at ‘...generating evidence-based data to inform policy on the health status of the population...’ but it seems the study population is men only? Or is it that for the analysis reported in this paper, the authors restricted their analysis to men (as it was investigating male perpetration of violence)? It would be helpful to first describe the AMBSO study and then how this cross-sectional analysis fits in. It seems that the authors used baseline data from the cohort study. I assume the 80.4% response rate was for the main study that recruited 2,014 men? Then of the 2,014 men recruited into the cohort, 1314 (65.2%) were included in this analysis. A flow diagram might be helpful. Characteristics of non-responders – age is referred to in the text but not reported in the supplementary table. It would be helpful to report in the text whether there are differences in the characteristics between responders and non-responders. Currently, the authors report only on the non-responders. The authors describe how HIV status was determined but prevalence of HIV was not an aim of this study and is not reported in the results. What is the justification for only measuring lifetime perpetration of IPV and not past 12 months as well? It seems strange as food insecurity was measured in past 12 months. Food insecurity measurement – it would be helpful to present how this was measured and categorised into none, low and high in a table. As this is main focus of the paper, it should be presented in the main text rather than as a supplementary table. Covariates – please provide justification for the choice of covariates.
--	---

	It is not clear how variables were selected for inclusion in the final model. Results Table 5 includes a range of factors – in addition to food insecurity and HIV-related factors mentioned in the title. Perhaps the title should be amended. Discussion Studies of male perpetration of violence in other countries have reported higher prevalence compared with the findings of this study. In addition, the Uganda DHS found much higher prevalence of physical violence than reported in this study. It would be helpful to discuss possible reasons for these differences. The authors suggest some reasons such as cultural, societal and institutional norms and values that vary across contexts but do not explain how and in what way. How does prevalence of male perpetration of violence reported in this study compare to reports from studies of women’s reports of violence in Uganda? The authors suggest that their findings complement existing literature on the experiences of IPV among women in Uganda but it is not clear in what way. The authors suggest that in the context of the SARS Cov-2 pandemic that food insecurity could be underlying risk factor for violence, but the evidence indicates that food insecurity is an existing risk factor. The concern is that the pandemic will lead to a considerable increase in the number of people facing food insecurity, leading to increases in violence against women. It would be helpful to include some discussion on interventions to address food insecurity targeted at men and women. Evidence suggests that these need to be combined with gender transformative interventions. Apart from age, there is no discussion about the other factors examined and their association with IPV, and how these findings compare with previous studies. There needs to be more discussion of the limitations of the study, particularly the possibility of under-reporting of physical and sexual violence. The authors note that variables such as income levels or employment status were not analysed but do not explain the reason, and if there were other important factors. Although the authors conducted a multivariable analysis, the possibility of residual confounding remains.
--	---

REVIEWER	Kathryn Falb IRC, USA
REVIEW RETURNED	03-Dec-2020

GENERAL COMMENTS	Overall the article was written and provides needed evidence on a burgeoning area of research examining the roles of food insecurity and IPV perpetration. I look forward to future research on better understanding and testing pathways between this relationship. I
--

	propose the following suggestions below to help strengthen and clarify the manuscript. Abstract In the introduction of the abstract, please clarify whether the prevention and response policies are in relation to IPV, food security, or both. Introduction Line 23; change intersection to intersecting The study (#23) which found correlates between hunger and alcohol consumption is interesting, but could be further detailed in the introduction as it seems there are different pathways between hunger and IPV and alcohol and IPV. Could the authors clarify this? There is also a small body of evidence from Cote d'Ivoire on food insecurity and IPV (from women's reports) the authors may want to refer to (e.g., Fong, et al, 2016). Methods The 321 men who opted out of responding to male IPV perpetration questions is concerning. It is appreciated more information is found in the supplemental tables. Could you describe whether there are food security differences, in addition to demographics, between men who completed this module and men who did not? How do the authors think this could bias the findings? I'm curious to know why the authors coded IPV the way that they did – do they hypothesize that the relationship between food insecurity and IPV is different between physical and sexual IPV perpetration? Given how the outcome is constructed it must be assumed that the authors would posit the relationships are different between these forms of violence. This explanation should be in the introduction. Regardless, I would also strongly recommend having a physical and/or sexual IPV variable outcome in addition to what is already presented in a logistic model for ease of comparison to other studies who typically use this modeling approach. Additionally, do the authors have any insights into severity or frequency or violence to further build nuanced understanding of this association? Can the authors also explain why lifetime IPV was chosen rather than past year (as I assume they also have this data)? It seems that although this study is cross-sectional in nature and causality cannot be ensured, that switching from a lifetime to past year timeframe for the outcome would help bolster the findings. I would recommend moving the information on assent for boys 13-17 to the ethical considerations section, rather than in the description of the sample. In the analyses, please also explain how the sampling was accounted for in the modeling. Was it that all males aged 13-80 in the communities were asked to participate and all households were asked to participate in the study? Were more than one male in a home allowed to be part of the study? If so, how does the analysis account for the intrahousehold correlation? Were any community-level correlations or weighting included in the modeling? Please provide more information on stratification and sampling- and how the sampling design was accounted for in the modeling. Please also add more information on data collection – was in electronic? Male enumerators only? Results
--	---

	Please include p-values in the tables, where appropriate (such as Table 4). In Table 4, age is now categorized into different groups, whereas Table 2 presents it continuously. Given the very large range 13-80 years, please potentially consider adding more categories for the 35+ group based on meaningful categories to better get a sense of this relationship. Also, please include the number of participants in the 13-24 years group in the table even if they serve as the reference group. This is particularly important as the Figure 1 on age and IPV shows a very clear relationships that is varying by different age groups. In Table 5, it seems the authors might be missing the 35+ age group in the results and/or models. Discussion Can the authors include more information about how they interpret the strength of the association between food insecurity and IPV in comparison to the other variables that were included (such as alcohol use which had a stronger relationship)? How do the authors conceptualize these other variables in relation to food insecurity and IPV– are they mediators? Confounders? Were there also any differences based on rural, urban or semi-urban areas? Please again include the high level of non-response, and the significant differences between men who responded and those that did not the IPV modules and how that might influence findings in the limitations.
--	--

VERSION 1 – AUTHOR RESPONSE

Reviewer comment	Response to comment
Reviewer 1	
General comment	
The manuscript needs a thorough proof-read and copy-edit as there are a number of typographical errors, e.g. should be temporal not temporary (3rd bullet in the strengths and limitations section).	The manuscript has been proof-read and frequent errors corrected.
Abstract	
The authors state that the aim of the study is to determine lifetime prevalence of male perpetrated IPV, and to assess its association with food insecurity, but it seems that the investigators were looking at a range of factors, such as age, education etc. This needs to be made clear.	We explained in the introduction that while we assess whether IPV perpetration is associated with food insecurity, the statistical analysis must also control for confounding from sociodemographic factors and HIV-risk behaviours such as alcohol and illicit drug use which have also been frequently associated with IPV. The justification for choosing these variables has been further clarified.
In the conclusion the authors state that food insecurity could worsen with the SARS-CoV-2 outbreak – I would suggest	Thank you for this comment. We have added this in the introduction.

adding to the introduction section that the prevalence of food insecurity has increased considerably in Uganda, which is likely to worsen with the SARS-CoV-2 outbreak.	
In the results, the authors state that 25-34-year-olds reported the highest prevalence of both physical and sexual violence – it would be helpful to include the prevalence for each form of violence.	This information is available in the results section but we think it is less relevant to include it in the abstract where we are actually giving room for the main findings.
In the methods the authors state that food insecurity was categorized as no, low and high but there is no reference to these categories in the results section.	The prevalence of different levels of food insecurity has been included in the results section. Please, see page 2.
The authors state in the results that there was no association between IPV perpetration and age but in the conclusion state that perpetration varied by age.	Thank you for this important observation. It is true that our findings show that the prevalence of IPV perpetration varies by age group. We have corrected this error by including some edits in this result. The conclusion is however unchanged.
Introduction	
The authors state that there are no previous studies examining the association between food insecurity and male perpetrated IPV in sub-Saharan settings, however, there are some studies such as Gibbs et al (2018) that have examined associations between poverty (including food insecurity) and IPV (including male perpetration). It would be better to state that there are few studies/limited research.	The study of Gibbs et al has been referenced in the background (page 6). We now also clarify that there are a few (rather than no) studies on the topic from sub-Saharan Africa
The aim needs to be stated more clearly. In addition to measuring prevalence of male perpetrated IPV, the authors investigated the association of a range of factors with IPV, not just food insecurity, as stated.	Thank you for this comment. The aim has been revised to read "... The current study aims to determine the prevalence of lifetime male IPV perpetration among a representative sample of males in two Ugandan districts, and to assess its association with food insecurity, and a range of social-demographics and health-risk behaviours".
Methods	
The description of this study/cross-sectional analysis in relation the AMBSO cohort study is slightly confusing. The authors state the AMBSO cohort study is aimed at ‘...generating evidence-based data to inform policy on the health status of the population...’ but it seems the study	Thank you for giving us the opportunity to clarify this section. The AMBSO cohort includes a broader population, however, for purposes of the current study the sample was restricted to men. Please see page 8.

population is men only? Or is it that for the analysis reported in this paper, the authors restricted their analysis to men (as it was investigating male perpetration of violence)? It would be helpful to first describe the AMBSO study and then how this cross-sectional analysis fits in. It seems that the authors used baseline data from the cohort study.	
I assume the 80.4% response rate was for the main study that recruited 2,014 men? Then of the 2,014 men recruited into the cohort, 1314 (65.2%) were included in this analysis. A flow diagram might be helpful.	The 80.4% response rate is for those who met the inclusion criteria (ever had a sexual relationship) i.e. 1635 men. A flow diagram has now been added to illustrate the selection process (Figure 1)
Characteristics of non-responders – age is referred to in the text but not reported in the supplementary table. It would be helpful to report in the text whether there are differences in the characteristics between responders and non-responders. Currently, the authors report only on the non-responders.	Thank you for the observation. The age (mean, SD), interquartile range (years) was indeed presented and compared for responders and non-responders in the supplementary table. Please see supplementary table 1 Although we referred to non-respondents in the text, we indicated that it was in comparison with the respondents. To make this clearer, we have rephrased the sentence (please, see pages 9-10)
The authors describe how HIV status was determined but prevalence of HIV was not an aim of this study and is not reported in the results.	We feel that it would be interesting to show the prevalence of HIV in our study population, given the interaction between HIV status and intimate partner violence- Since HIV testing is part of the PHS cohort's routine data collection, presenting this data gives readers more information about these intersecting burdens in the communities under study. In the Results section, Table 3-shows respondent characteristics, the HIV prevalence was presented as one of several background characteristics.
What is the justification for only measuring lifetime perpetration of IPV and not past 12 months as well? It seems strange as food insecurity was measured in past 12 months.	Thank you for this important comment. We agree that capturing the 12-month prevalence of IPV would indeed have been important (given that this is a commonly used measure in the literature on VAW), but this variable was not captured as part of the standard PHS cohort questionnaire which focused on lifetime experiences of violence. We are using baseline data from an open population cohort, and our intention is to describe the burden of male IPV perpetration over their lifetime. Assumptions are, based on previous

	literature, that men who have perpetrated IPV at some point, are very likely to repeat this behavior. Thus, using lifetime prevalence of IPV captures more of men's violence potential in the community than if we had used recent (past-year) estimates. This justification has now been included in the manuscript on page 11 In terms of food insecurity, a 12-month recall period is standard and one of the most widely recommended time periods for assessing household food availability.
Food insecurity measurement – it would be helpful to present how this was measured and categorised into none, low and high in a table. As this is main focus of the paper, it should be presented in the main text rather than as a supplementary table.	Thank you. The supplementary table has been modified and included in the main manuscript body, showing the questions for assessment and how the coding was done during analysis
Covariates – please provide justification for the choice of covariates.	We have included a justification for including the number of sexual partners, condom use and circumcision as covariates. The other covariates are sociodemographic factors, while binge alcohol use and illicit drugs consumption are known to be associated with IPV perpetration. (Please note that we are also mindful of keeping within the word limits for the manuscript)
It is not clear how variables were selected for inclusion in the final model.	We have explained this, indicating that we chose variables in the bivariate analysis that have a theoretical link to IPV to build the full regression model
Results	
Table 5 includes a range of factors – in addition to food insecurity and HIV-related factors mentioned in the title. Perhaps the title should be amended.	Thank you for this observation, we have adapted the Table in line with your recommendation.
Discussion	
Studies of male perpetration of violence in other countries have reported higher prevalence compared with the findings of this study. In addition, the Uganda DHS found much higher prevalence of physical violence than reported in this study. It	On page 23, we have included possible explanations for the lower prevalence obtained in this study.

would be helpful to discuss possible reasons for these differences. The authors suggest some reasons such as cultural, societal and institutional norms and values that vary across contexts but do not explain how and in what way.	As we found no concrete examples of cultural, societal and institutional norms and how they influence our findings, we decided to remove this part from the discussion.
How does prevalence of male perpetration of violence reported in this study compare to reports from studies of women's reports of violence in Uganda? The authors suggest that their findings complement existing literature on the experiences of IPV among women in Uganda but it is not clear in what way.	The meaning of the sentence has been modified after careful thinking. "the findings of this study add to the existing literature on violence against women in Uganda using self-reports of perpetration"
The authors suggest that in the context of the SARS Cov-2 pandemic that food insecurity could be underlying risk factor for violence, but the evidence indicates that food insecurity is an existing risk factor. The concern is that the pandemic will lead to a considerable increase in the number of people facing food insecurity, leading to increases in violence against women	This has been corrected, indicating that "In the present context of SARS CoV2 pandemic, food security could worsen, thereby potentially increasing the risk of male IPV perpetration"
It would be helpful to include some discussion on interventions to address food insecurity targeted at men and women. Evidence suggests that these need to be combined with gender transformative interventions.	These interventions have been discussed (please, see page 26).
Apart from age, there is no discussion about the other factors examined and their association with IPV, and how these findings compare with previous studies.	It was actually discussed that "After adjusting for food insecurity and sociodemographics, the study concluded that men who consumed alcohol in the past year and have multiple sexual partners had a significantly higher risk of perpetrating IPV, which is in line with other studies". Please, see page 26
There needs to be more discussion of the limitations of the study, particularly the possibility of under-reporting of physical and sexual violence.	We have further explained the potential of under-reporting in the discussion. Please, see page 24
The authors note that variables such as income levels or employment status were not analyzed but do not explain the reason, and if there were other important factors. Although the authors conducted a multivariable analysis, the possibility of residual confounding remains.	Although in this Cohort, we collect data on income status and employment levels, by the time of doing this analysis and writing this manuscript, general categorization of this data was not complete, and thus decided not include the analysis for this paper.

Reviewer 2	
Abstract	
In the introduction of the abstract, please clarify whether the prevention and response policies are in relation to IPV, food security, or both.	This has been made clearer on page 2
Introduction	
Line 23; change intersection to intersecting	We have corrected “intersection” to “intersecting”
The study (#23) which found correlates between hunger and alcohol consumption is interesting, but could be further detailed in the introduction as it seems there are different pathways between hunger and IPV and alcohol and IPV. Could the authors clarify this?	More details have been added. They include information on other associations that were found, not only for perpetration of violence but also violence experiences. Poor mental status (sadness) was also associated but not analyzed as a mediator of IPV in the paper.
There is also a small body of evidence from Cote d’Ivoire on food insecurity and IPV (from women’s reports) the authors may want to refer to (e.g., Fong, et al, 2016).	Thank you very much for this suggestion, we found this article to be very relevant and have included it in the Introduction. Please, see page 6.
Methods	
The 321 men who opted out of responding to male IPV perpetration questions is concerning. It is appreciated more information is found in the supplemental tables. Could you describe whether there are food security differences, in addition to demographics, between men who completed this module and men who did not? How do the authors think this could bias the findings?	We have included in online supplementary appendix 1 showing the comparison of food insecurity among the responder and non-responder. The analysis showed that there was no significant difference in the food insecurity in the two groups.
I’m curious to know why the authors coded IPV the way that they did – do they hypothesize that the relationship between food insecurity and IPV is different between physical and sexual IPV perpetration? Given how the outcome is constructed it must be assumed that the authors would posit the relationships are different between these forms of violence. This explanation should be in the introduction. Regardless, I would also strongly recommend having a physical and/or sexual IPV variable outcome in	 • Our coding of the IPV was based on the analytical approach used in the similar WHO multi-country survey of male perpetration of IPV and its associated factors in Asia and the Pacific (Fulu et al, Lancet 2013)-although food insecurity was not one of the covariates explored in that study. This reference is given in the methods section and builds on the well-established practice within the field of violence against women research to separate different forms of violence (physical, sexual, psychological). • Yes, our assumption is that the associations of food insecurity are different for different typologies of violence, similar to how Fulu et al

addition to what is already presented in a logistic model for ease of comparison to other studies who typically use this modeling approach. Additionally, do the authors have any insights into severity or frequency or violence to further build nuanced understanding of this association?	analyzed factors associated with IPV perpetration by males. This has been factored in the analysis section (not the introduction section as you suggested).  • Unfortunately, due to space constraints in the survey, this study did not collect data on frequency or severity of IPV perpetration. This will be an important area for future studies.
Can the authors also explain why lifetime IPV was chosen rather than past year (as I assume, they also have this data)? It seems that although this study is cross-sectional in nature and causality cannot be ensured, that switching from a lifetime to past year timeframe for the outcome would help bolster the findings.	Thank you for this important comment. We agree that capturing the 12-month prevalence of IPV would indeed have been important (given that this is a commonly used measure in the literature on VAW), but this variable was not captured as part of the standard PHS cohort questionnaire which focused on lifetime experiences of violence. We are using baseline data from an open population cohort, and our intention is to describe the burden of male IPV perpetration over their lifetime. Assumptions are, based on previous literature, that men who have perpetrated IPV at some point, are very likely to repeat this behavior. Thus, using lifetime prevalence of IPV captures more of men's violence potential in the community than if we had used recent (past-year) estimates. This justification has now been included in the manuscript on page 11
I would recommend moving the information on assent for boys 13-17 to the ethical considerations section, rather than in the description of the sample.	This section was removed and reported under ethical considerations.
In the analyses, please also explain how the sampling was accounted for in the modeling. Was it that all males aged 13-80 in the communities were asked to participate and all households were asked to participate in the study? Was more than one male in a home allowed to be part of the study? If so, how does the analysis account for the intrahousehold correlation? Were any community-level correlations or weighting included in the modeling? Please provide more information on stratification and	Yes, every eligible male 13-80 years in the study community who consented to participate was included. We have highlighted in the manuscript section "study population and sampling" that "ALL males 13-80 years in the study communities and households" was the study population. All the eligible males in each household were included from each community. We have clarified that no weighting was included in the analysis.

sampling- and how the sampling design was accounted for in the modeling.	
Please also add more information on data collection – was in electronic? Male enumerators only?	This has been added. Thank you!
Results	
Please include p-values in the tables, where appropriate (such as Table 4).	P values have been included in (now) table 5
In Table 4, age is now categorized into different groups, whereas Table 2 presents it continuously. Given the very large range 13-80 years, please potentially consider adding more categories for the 35+ group based on meaningful categories to better get a sense of this relationship. Also, please include the number of participants in the 13-24 years group in the table even if they serve as the reference group. This is particularly important as the Figure 1 on age and IPV shows a very clear relationships that is varying by different age groups	 • Although we have 522 participants in the 35+ age group, the prevalence of “NO IPV” alone is highest (72.4%), compared to lower age groups. Creating more age groups from 35+ generates very small cell numbers which make modelling iterations futile. • The number and proportions of participants in the reference groups have been added.
In Table 5, it seems the authors might be missing the 35+ age group in the results and/or models	Now table 6. The table has been updated with results for the 35+ years age group. Thank you!
Discussion	
Can the authors include more information about how they interpret the strength of the association between food insecurity and IPV in comparison to the other variables that were included (such as alcohol use which had a stronger relationship)? How do the authors conceptualize these other variables in relation to food insecurity and IPV– are they mediators? Confounders?	“Although an association was found between food insecurity and IPV perpetration, literature has not demonstrated a direct pathway between food insecurity and IPV perpetration. Mediating factors which have been examined through structural equation models include poor mental health, gender attitudes, controlling behaviours and alcohol consumption. In interpreting the association between food insecurity and IPV perpetration, factors such as alcohol (although shown to have some direct effects on IPV perpetration,(24)), and having multiple sexual partners mostly fall within the ranks of mediators, as illustrated by literature”.
Were there also any differences based on rural, urban or semi-urban areas? Please again include the high level of non-response, and the significant differences between men who responded and those	Due to small sample, analyzing the differences between communities did not attract the interest of the current study. The comparison was rather made at the district level with a larger sample (table 5), in which no significant difference in

that did not the IPV modules and how that might influence findings in the limitations	perpetration of physical and sexual IPV was observed between Hoima and Wakiso. We have explained in the discussion, under limitations, that a good number of men did not respond to the baseline IPV questions. The sensitivity analysis points that this had no effect on the findings.

VERSION 2 – REVIEW

REVIEWER	Sheila Harvey London School of Hygiene & Tropical Medicine UK
REVIEW RETURNED	23-Feb-2021

GENERAL COMMENTS	General comment The manuscript requires further proof-reading and copy-editing. I noticed spelling and grammatical errors throughout the manuscript. Aim Although the authors have added that they looked at a range of other factors, it is obvious that the authors main interest is the association between food insecurity and IPV – this should be made clear that this is the primary focus of the paper, and the association between IPV and other factors were secondary outcomes. Methods More detail is needed on the sample size calculation – the authors state that “a sample of 738 males was needed to estimate the associations in the current study with 80% at the alpha=0.05 level.” It needs to be stated which associations they are referring to and what assumptions have been made. The authors refer to food insecurity as the independent variable, and all other variables (e.g. alcohol use) as covariates, however, assessing the association between these variables and IPV is part of the aim of the study. Was any adjustment made for multiple comparisons? Results In the text reference to some of the tables seems to be incorrect. For example, in the paragraph entitled Multivariable analysis, the authors refer to Table 5 but I think they mean Table 6. I think the titles of Tables 5 and 6 need to be revised as they are reporting risk ratios for a number of factors, not just food insecurity.
---

	Can a footnote be added to Table 6 to indicate which variables were adjusted for in the analysis? It would also be helpful to add in the p values so readers can assess the strength of evidence. Discussion Much of the discussion focusses on the association between food insecurity and IPV perpetration, with little discussion of the other factors examined. Food insecurity is an indicator of extreme poverty, and I think there should be more discussion of the literature on poverty as direct driver and indirect driver of IPV. The authors recommend a number of interventions to address food insecurity given the association between food insecurity and perpetration of IPV. However, they comment that other studies have not demonstrated a direct pathway between food insecurity and IPV. Furthermore, they note that because this was a cross-sectional study a temporal relationship cannot be established. So, it is unclear how or whether the interventions recommended are likely to have an impact. The authors also mention other interventions such as SASA! and the Program P initiative, but it is not clear how these fit in with the findings of this study, which is limited by its cross-sectional design.
--	---

REVIEWER	Kathryn Falb International Rescue Committee
REVIEW RETURNED	14-Feb-2021

GENERAL COMMENTS	Overall, the authors have incorporated suggested revisions into the manuscript and is suitable for publication.
---

VERSION 2 – AUTHOR RESPONSE

Reviewer comment	Response to comment
Reviewer 1	
General comment	
The manuscript requires further proof-reading and copy-editing. I noticed spelling and grammatical errors throughout the manuscript.	The manuscript has been further proof-read and copy-edited.
Aim	
Although the authors have added that they looked at a range of other factors, it is obvious that the authors main interest is the association between food insecurity and IPV – this should be made clear that	We agree that investigating the association between food insecurity and IPV was a key focus of our study. The primary aim, “ to determine the prevalence of lifetime male IPV perpetration....., and to assess its association

this is the primary focus of the paper, and the association between IPV and other factors were secondary outcomes.	with food insecurity", has been clarified. The secondary aim "to determine whether the strength of the association between food insecurity and male IPV perpetration is affected by social-demographic factors and health-risk behaviours in the districts" has also been clarified. Please see page 6 for these corrections.
Methods	
More detail is needed on the sample size calculation – the authors state that “a sample of 738 males was needed to estimate the associations in the current study with 80% at the alpha=0.05 level.” It needs to be stated which associations they are referring to and what assumptions have been made.	More details about the sample size estimation have been provided to clarify the assumptions and the various inputs that guided the calculation of the study sample. Please refer to page 8 for the changes.
The authors refer to food insecurity as the independent variable, and all other variables (e.g. alcohol use) as covariates, however, assessing the association between these variables and IPV is part of the aim of the study.	Thank you for pointing out this mistake. We have clarified that food insecurity, socio-demographic and health risk behaviours were the independent variables. Please see page 12.
Was any adjustment made for multiple comparisons?	This has been clarified on page 13. The adjusted variables are shown as a footnote under table 6.
Results	
In the text reference to some of the tables seems to be incorrect. For example, in the paragraph entitled Multivariable analysis, the authors refer to Table 5 but I think they mean Table 6.	We have cross-checked and found that the table was correctly cited in the text. Please see page 20.
I think the titles of Tables 5 and 6 need to be revised as they are reporting risk ratios for a number of factors, not just food insecurity.	Thank you. The table titles have been revised accordingly.
Can a footnote be added to Table 6 to indicate which variables were adjusted for in the analysis? It would also be helpful to add in the p values so readers can assess the strength of evidence.	A footnote has been added and the p-values included in table 6.
Discussion	

Much of the discussion focusses on the association between food insecurity and IPV perpetration, with little discussion of the other factors examined.	Alcohol use, multiple sexual partners and the use of illicit drugs which were significantly associated with IPV have been further discussed. More of the discussion is now dedicated to the association between food insecurity and IPV because it was the primary focus of the study, while being cautious to not exceed the specified word count.
Food insecurity is an indicator of extreme poverty, and I think there should be more discussion of the literature on poverty as direct driver and indirect driver of IPV.	We entirely agree and have added more on what is known about the relationship between financial hardship and IPV in the introduction (see page 5). It would of course have been ideal if we had complete data on socio-economic status (i.e. income levels, housing etc.) but an index indicating socioeconomic status at household level using the original census assessment is under construction and waiting for that would have delayed this publication that we think conveys an important message that needs to come out. We acknowledge this limitation of the study.
The authors recommend a number of interventions to address food insecurity given the association between food insecurity and perpetration of IPV. However, they comment that other studies have not demonstrated a direct pathway between food insecurity and IPV. Furthermore, they note that because this was a cross-sectional study a temporal relationship cannot be established. So, it is unclear how or whether the interventions recommended are likely to have an impact.	It is true that a temporal association between IPV and food insecurity has not been established. But we opine that the findings of this study should serve as an alert to existing interventions and potentially new initiatives to address IPV in particular in light of a dramatic increase in both IPV and food insecurity reports as a consequence of the COVID-19 pandemic. The surveillance design of this so far rather new population-based cohort in Uganda will enable longitudinal surveys to investigate the longer-term development of food insecurity (as well as IPV) and drivers of food insecurity, which could throw more light onto the topic and improve the quality of the evidence. Please see page 25 for the edits that further clarify the recommendations based on our evidence.
The authors also mention other interventions such as SASA! and the Program P initiative, but it is not clear how these fit in with the findings of this study, which is limited by its cross-sectional design.	Thank you for pointing this out. Considering the consistency of our findings with findings from other studies that link food insecurity and violence against women, we believe it is worthy to alert the SASA and Program P initiatives on the need to strengthen and scale up their activities and consider advocating for food security measures as part of the packages to assess and respond to IPV. This is now further highlighted in the Discussion section on page 27